Exceptional soft tissues preservation in a mummified frog-eating Eocene salamander

Tissier Jérémy jeremy.tissier@unifr.ch 1 2
Rage Jean-Claude 3
Laurin Michel 3
1 Cenozoic Research Group, JURASSICA Museum , Porrentruy , Switzerland
2 Department of Geosciences, University of Fribourg , Fribourg , Switzerland
3 Département Histoire de la Terre, UMR 7207, Centre de Recherches sur la Paléobiodiversité et les Paléoenvironnements, CNRS/MNHN/UPMC (Sorbonne Universités), Museum national d’Histoire naturelle , Paris , France
Abdala Virginia
Electronic publication date: 2017 Oct 3
Publication date: 2017
Volume: 5
Electronic Location ID: e3861
Received 2017 Aug 22; Accepted 2017 Sep 7
Copyright: ©2017 Tissier et al.
Copyright year: 2017
Copyright holder: Tissier et al.
License: This is an open access article distributed under the terms of the Creative Commons Attribution License, which permits unrestricted use, distribution, reproduction and adaptation in any medium and for any purpose provided that it is properly attributed. For attribution, the original author(s), title, publication source (PeerJ) and either DOI or URL of the article must be cited.
License URL: https://creativecommons.org/licenses/by/4.0/

Keywords: Soft tissues, Phosphorites du Quercy, Exceptional preservation, Extinct urodele, Ecology, Synchrotron microtomography, Three-dimensional preservation, Eocene

Funding: CNRS the French Ministry of Research, and Sorbonne Universités Swiss National Science Foundation 200021_162359 Jérémy Tissier, Jean-Claude Rage, and Michel Laurin were funded by recurring grants from the CNRS, the French Ministry of Research, and Sorbonne Universités to the CR2P. Jérémy Tissier is currently financially supported by the Swiss National Science Foundation (project 200021_162359). The funders had no role in study design, data collection and analysis, decision to publish, or preparation of the manuscript.

==============================
Fossils are almost always represented by hard tissues but we present here the exceptional case of a three-dimensionally preserved specimen that was ‘mummified’ (likely between 40 and 34 million years ago) in a terrestrial karstic environment. This fossil is the incomplete body of a salamander, Phosphotriton sigei, whose skeleton and external morphology are well preserved, as revealed by phase-contrast synchrotron X-ray microtomography. In addition, internal structures composed of soft tissues preserved in three dimensions are now identified: a lung, the spinal cord, a lumbosacral plexus, the digestive tract, muscles and urogenital organs that may be cloacal glands. These are among the oldest known cases of three-dimensional preservation of these organs in vertebrates and shed light on the ecology of this salamander. Indeed, the digestive tract contains remains of a frog, which represents the only known case of an extinct salamander that fed on a frog, an extremely rare type of predation in extant salamanders. These new data improve our scarce knowledge on soft tissue anatomy of early urodeles and should prove useful for future biologists and palaeontologists working on urodele evolutionary biology. We also suggest that the presence of bat guano and carcasses represented a close source of phosphorus, favouring preservation of soft tissues. Bone microanatomy indicates that P. sigei was likely amphibious or terrestrial, and was probably not neotenic.

Introduction

The ‘Phosphorites du Quercy’, in southwestern France, include numerous karstic fissures in-filled by phosphatic sediments rich in vertebrate remains (Legendre et al., 1997; Pélissié & Sigé, 2006). Almost all remains appear as classical disarticulated fossil bones, but a few of them (a salamander, anurans and snakes) are spectacular cases of exceptional preservation; the animals are entirely mineralized, including the skin, in three dimensions. Unfortunately, these ‘mummies’ were collected in the 19th century and their precise provenance and geological age are unknown. However, it is suspected that they come from the late middle or late Eocene (Laloy et al., 2013; Tissier et al., 2016).

Until recently, only the external morphology of the ‘mummies’ was known. However, recent tomographic studies showed that the skeleton is preserved within the ‘mummies’ of the frog Thaumastosaurus gezei (Laloy et al., 2013) and of the salamander Phosphotriton sigei) (Tissier et al., 2016). The specimen of P. sigei includes a large part of the trunk (preserved posterior to the shoulder girdle), the anterior portion of the tail and the proximal portions of the hind limbs (Fig. 1A). The right side of the trunk is crushed. Diagnostic external features include the absence of scales, the presence of costal grooves visible on the left side, and the presence of a longitudinally slit-shaped cloaca.

Figure 1 Specimen MNHN.F.QU17755, holotype of Phosphotriton sigei.

(A and B) Fossil in dorsal and ventral views. Some characteristics of urodeles, such as costal grooves or scaleless skin, are observable on the external aspect of the specimen. The cloaca and vertebral column are visible. The dotted line represents the position of the tomogram illustrated in Fig. 1C. (C) Tomogram of the tail part of the animal showing the muscles, in green, ventral and lateral to the vertebrae, and the spinal cord preserved inside the neural canal of a vertebra. Bony material is characterized by a dark grey shade, because of its light density, compared to the mineral matrix (grey or white) and void (black). Soft-tissues are also mostly darker than the mineral matrix, but are mainly recognizable by their structure and shape, on tomograms or in 3D. (D) 3D reconstruction of undetermined tail muscles, in green, which could attach to the ischium or femur. Dotted line represents the position of the tomogram illustrated in Fig. 1C.

The study of the skeleton of Phosphotriton confirmed that this fossil is a urodele amphibian; more precisely, the phylogenetic analysis presented by Tissier et al. (2016) suggested that it is a stem-salamandrid, although they did not definitely discard relationships with the Plethodontidae.

The microtomography of Phosphotriton clearly suggested also that, in addition to the skeleton, soft tissues were preserved. Subsequent segmentations indeed displayed various soft tissues within this specimen, which are the subject of the present article. We show here that the observed organs are not infills of cavities but are really the organs themselves that were permineralized.

Materials and Methods

The only specimen of P. sigei (MNHN.F.QU17755) was investigated with the help of propagation phase contrast synchrotron X-ray microtomography, which gives a better contrast to differentiate tissues from the mineral matrix than traditional absorption based synchrotron X-ray microtomography. The method and parameters of acquisition are described in Tissier et al. (2016). A 3D model is given in Supplemental Information 1 in 3D PDF file format.

Several structures composed of soft tissues are preserved and may be identified on the tomograms. They can be distinguished from bones and mineral matrix by their shape, density on tomograms and structure. Their identification is based on comparisons with the literature on urodele soft anatomy because dissecting extant specimens would not have added to what may be drawn from the available literature. Therefore, we use their position in the body, their shape in three dimensions, and their internal structure on tomograms to identify them based on comparisons with existing descriptions. Some of them remain difficult to identify precisely, for several reasons (incompleteness of the organ, segmentation difficulties, small size, etc.). Proposed identifications are therefore tentative in some cases (e.g., an organ of the uro-genital system), although some appear to be certain (spinal cord, lumbosacral plexus).

To assess the lifestyle of P. sigei, we analysed the compactness profile of femoral mid-diaphyseal virtual cross-sections. We then used these data to infer the lifestyle with the inference models published by Laurin, Canoville & Quilhac (2009). These are based on statistical analyses of femoral compactness profiles of 46 extant urodele species. Variables in the models were selected through backward elimination and forward selection procedures, respectively, which led to two models with different combinations of variables.

Results

Muscles. Not all muscles appear to have been fossilized. In addition, most muscles were not segmented, because of their irregular, ill-defined contour; their segmentation would have required too much subjective interpretation and would have been very time-consuming. It has not been possible to precisely identify the preserved muscles, as their position in the specimen is not sufficient for this. Only three of them were segmented: they are recognizable by their fibrous structure and shape (Figs. 1C–1D). We suppose that these may be three ventral caudal muscles described by Francis (1934: 102–103), which arise from the fourth caudal vertebra (i.e., M. caudali-pubo-ischio-tibialis, M. ischio-caudalis (the most mesial one, which inserts on the posterior border of the ischium) and M. caudalifemoralis (the most lateral one, which inserts on the femur)). Francis (1934) described them as having an ‘oval cross-section’, and being ‘narrow and strap-like’, which fits the muscles disclosed here. Their function is to flex the tail.

Figure 2 Tomogram of the trunk portion of the specimen MNHN.F.QU17755.

Spinal cord is in blue, within the neural canal of a trunk vertebra (in yellow).

Spinal cord. It is preserved and visible in section in some vertebrae, inside the neural canal (Fig. 1C, 2). In the vertebrae where it is not preserved, only an empty space is visible (black on the tomogram). Unfortunately, that organ could not be segmented because its preservation is too uneven. No bony support of the spinal cord is visible. Spinal cord supports are bony processes that extend in the neural canal of vertebrae (Wake & Lawson, 1973; Skutschas, 2009; Skutschas & Baleeva, 2012). The fact that supports do not appear on the images does not necessarily mean that they were absent. These structures, which occur in various salamanders, are tiny and difficult to detect on tomograms (Skutschas & Baleeva, 2012; PP Skutschas, pers. comm., 2015).

It seems clear that in this specimen, the soft tissues are mineralized, even internally, and do not represent cavity filling. Indeed, the structure of the spinal cord is in some rare places well preserved, in three dimensions. Notably, the external surface of the cord is bordered by empty space on tomograms (Fig. 2), which would not happen if this was a case of cavity filling preservation. This ‘empty space’ was originally occupied by the cerebrospinal fluid, which cannot fossilize. Internal structure is difficult to discern but it is nevertheless reminiscent to what can be observed in extant urodeles, with a central canal (see Davis, Duffy & Simpson, 1989: fig. 6A for example).

Lumbosacral plexus. This plexus comprises three nerves that emerge from the spinal cord through the spinal foramina of the last trunk vertebra, the sacral vertebra and the first caudosacral vertebra. These spinal foramina are large (Tissier et al., 2016: figs. 5B and 6B–6C). These three nerves merged lateral to the ilia to form the lumbosacral plexus (Figs. 3A–3B) and the resulting nerve entered the hind limb; this is similar to the disposition observed in Necturus by Wischnitzer (1979). The nerve exiting the last trunk vertebra corresponds to the ‘sixteenth spinal nerve’ in Salamandra Francis (1934: 173). The middle nerve of the plexus, emerging from the sacral vertebra, is the thickest, correlatively with the size of the foramen. It is termed ‘seventeenth spinal nerve’ in Salamandra by Francis (1934). The nerve exiting from the first caudosacral vertebra, called nervus spinalis 18 in Salamandra (Francis, 1934), is very thin and the preserved part does not meet the other nerves of the plexus, which are much thicker. However, in view of its orientation, we presume that it took part in the plexus and that the missing part results from incomplete fossilization or from an insufficient contrast on tomograms, leading to segmentation artefacts.

Figure 3 Exceptional preservation of nerves, digestive tract and stomachal content.

(A and B) 3D reconstructions of the pelvic section of MNHN.F.QU17755, in laterodorsal (A) and ventral (B) views. The lumbosacral plexus (in blue) is partly preserved. Nerves exit the last trunk, the sacral and the first caudosacral vertebrae through the spinal nerve foramina. (C) Preserved bones of an anuran frog (ranoid?), in green, inside the digestive tract (not shown, to better reveal its content; see Fig. 3F) of MNHN.F.QU17755. (D) Anuran humerus found inside digestive tract of MNHN.F.QU17755, in lateral and ventral views. (E) Anuran vertebrae found inside digestive tract of MNHN.F.QU17755. The centrum is very thin; the holes may represent segmentation artifacts. (F) 3D reconstruction of MNHN.F.QU17755 in ventral view, showing the nearly complete digestive tract. The caudal end is very close to the cloaca, and is bordered near the pelvic girdle by presumed dorsal cloacal glands (see Fig. 4A). The dotted line represents the position of the virtual section illustrated in Fig. 3G. (G) Virtual section of the trunk, showing the digestive tract (in yellow) and its content (frog bones).

Figure 4 Exceptional preservation of cloacal glands (?) and lung.

(A) 3D reconstruction of supposed dorsal and ventral cloacal glands, in ventral view, under the two ischia (not shown). The dorsal cloacal glands are located between the first and second caudosacral vertebrae and the digestive tract (see Fig. 4B). The ventral cloacal glands are located under the digestive tract and anterodorsal to the cloaca. The dotted line represents the position of the virtual section illustrated in Fig. 4B. (B) Virtual section of the pelvic girdle, illustrating the digestive tract and the dorsal cloacal glands, between a caudal vertebra and the two ischia. (C) 3D reconstruction of the incomplete lung (in blue), inside the specimen MNHN.F.QU17755, in oblique anterior view. It is located lateroventrally to the trunk vertebrae, in the anteriormost preserved part of the fossil. The dotted line represents the position of the tomogram illustrated in Fig. 4D. (D) Virtual section of the anteriormost preserved part of MNHN.F.QU17755, showing the inside of the lung in lateral view.

Digestive system. The alimentary canal is particularly easy to identify by its circular outline on the tomograms in transverse section. It is visible in most of the specimen length, up to the level of the pelvic girdle. It is very well preserved and its shape in three dimensions leaves little to no doubt about its identification (Figs. 3F–3G). Its diameter is quite variable and no well-defined stomach may be discerned, which is a characteristic of various urodeles (Delsol, Flatin & Exbrayat, 1995).

Here, the content of the digestive system is preserved (Figs. 3C–3E), a very rare and exceptional phenomenon: a few bones are present in the digestive tract, including a small humerus (five mm long) of an undetermined anuran, recognizable by its typical spherical distal articular condyle. Four vertebrae in connection are also present and could belong to that same young anuran.

Urogenital organ. Two paired organs are located just posterior to the pelvic girdle: one ventral to the first two caudosacral vertebrae, the other ventral to the second and third caudosacral vertebrae and dorsal to the cloaca (Figs. 4A–4B). Each is comprised of two elongate, fusiform elements situated on both sides of the cloaca. On the specimen, the cloaca is an elongate slit located just posterior to the hind limbs (Figs. 1B, 3F and 4A). These paired organs are approximately five mm long. Both parts of the dorsal most organ, ventral to the first two caudosacral vertebrae, are connected by a plate-like structure that is probably an artefact, given that it was difficult to differentiate it from the surrounding matrix and other elements during segmentation. The two parts of the ventral most organ are also connected, but it is very difficult to tell how, because of low contrast on tomograms. Assuming that these two organs are really paired, i.e., that the plate-like element is an artefact, the elongate parts may represent cloacal glands, the testicles, or the kidneys. In urodeles, testicles and kidneys may be similarly elongated (Delsol, Blond-Fayolle & Flatin, 1995; Gipouloux & Cambar, 1995), but they are located more cranially. These structures are thus more likely to represent dorsal and ventral cloacal glands, but this conclusion must remain tentative because the morphology of these glands in extant urodeles remains poorly described, though some histological descriptions have been published (Sever, 1981; Sever, 1992). According to Francis (1934), the male cloaca is surrounded by ‘a large tubular gland’, which fits the description of the ventral glands preserved here. These glands are not found in females Salamandra, which would mean that this fossil specimen was a male.

Lung. It was briefly described by Tissier et al. (2016), but a new description is given here, nevertheless. This organ is observable at the anterior part of the specimen, on the left side (Fig. 4C). The anterior portion is missing. The preserved part is triangular in dorsal or ventral view, its tip being directed caudally, and flattened in cross section (Fig. 4D). The section shows a vacuolar structure. Despite the absence of the anterior portion, the position of that organ in the body, ventral to the ribs (in the thoracic region), its shape and its vacuolar internal structure suggest that it is a lung (Francis, 1934; M Laurin, pers. obs., 2014). Within Caudata, the presence of a lung is primitive, but it remains useful to exclude some taxonomic affinities (i.e., within Plethodontidae).

Discussion

Ecology. The presence of anuran bones in the digestive tract of the fossil (Figs. 3C–3E) is evidence of a type of predation that is very rare in urodeles. Preying on frogs was reported in Amphiuma (Montaña, Ceneviva-Bastos & Schalk, 2014), a large and especially voracious extant urodele. Another voracious urodele, Necturus, has been reported (Hamilton, 1932) to have eaten other urodeles (Desmognatus and Eurycea), but not frogs. P. sigei was relatively small and the swallowed anuran, although small, was likely a metamorphosed individual, as shown by the well-shaped humeral condyle, but not a fully grown adult, as shown by the broad neural canal, assuming that the vertebrae belong to the same individual as the humerus. The straight diaphysis of the humerus and the position of the humeral condyle in line with the diaphysis suggest that the prey was a ranoid. Ranoids were already reported from the Phosphorites (Rage, 1984; Rage, 2016). The length of the humerus (five mm) suggests that the individual measured about 18–20 mm in snout-vent length.

To further investigate the ecology of the animal, we studied the microanatomy of the femur, through a transverse virtual section of the diaphysis on tomograms, and calculated its compactness profile with the software Bone Profiler (Girondot & Laurin, 2003). Without much surprise, both inference models (based on backward elimination and forward selection procedures, respectively) presented by Laurin, Canoville & Quilhac (2009) suggest an amphibious or terrestrial lifestyle (see Supplemental Information). This would suggest that P. sigei was not neotenic because all extant neotenic urodeles are strictly aquatic.

Exceptional preservation. The three-dimensionally preserved organs described here rank among the oldest known in vertebrates (even though the geological age of the studied fossil could only be determined indirectly). Putative lungs were described from the Devonian Bothriolepis (Denison, 1941), but this interpretation has recently been refuted by Goujet (2011). A probable ‘lung’ has also been observed in the actinistian sarcopterygian Axelrodichthys araripensis from the Cretaceous (Brito et al., 2010), but it is structurally very different from the regular lung of other vertebrates; it is geologically older than the lung of Phosphotriton sigei, but its fossilization is linked to the fact that it was originally mineralized (in vivo). The spinal cord, although we have not segmented it and it is not visible on all original virtual sections, is partly preserved. It is, to our knowledge, the only case of three-dimensional fossil preservation of that structure. The spinal cord is quite infrequent in the fossil record. It is known in the tadpoles of the Miocene frog Rana pueyoi; in fact, in the latter fossils, McNamara et al. (2010) described more precisely the nerve chord, which is the embryonic antecedent of the spinal cord. In these fossils, the cord is preserved in two dimensions. To our knowledge, the specimen of Phosphotriton is the only example of a fossilized nerve plexus in vertebrates. The three-dimensional preservation of the digestive tract documented here is also particularly exceptional. In fossils, this tract is generally two-dimensionally preserved, with even sometimes its content (Dal Sasso & Signore, 1998; McNamara et al., 2010), or the tract content may be preserved without impression of the tract itself (e.g., Piñeiro et al., 2012), but never to our knowledge have a three-dimensional fossilized tract and its content been reported in vertebrates; however, three-dimensional tracts, with perhaps remnants of the content, have recently been described in fossilized arthropods, which also come from the ‘Phosphorites du Quercy’ (Schwermann et al., 2016a). Phosphotriton may also be the only case of fossilization of an organ of the urogenital system (likely cloacal glands) among vertebrates (even though our interpretation of this structure is tentative) and it is the first known instance of an extinct salamander taxon and of a putative salamandrid (extinct or not) that fed on an adult anuran. Muscles reported here, on the contrary, are not the oldest known, as they have been reported in Eastmanosteus calliaspis, a Late Devonian placoderm (Trinajstic et al., 2007) and in the actinistian sarcopterygian Wenzia latimerae from the Late Oxfordian (Clément, 2005), for example.

This case of exceptional preservation is difficult to explain, more specifically as the fossiliferous locality that produced the fossil is unknown. It is suspected, but cannot be demonstrated, that all mummies from the ‘Phosphorites du Quercy’ come from a single, lost locality. It is striking that none of the numerous fossiliferous sites of the Phosphorites du Quercy investigated during the last five decades or so produced ‘mummies’. Equally strange is that none of the mummies pertain to Mammalia, as most skeletal remains found in the Phosphorites du Quercy are mammals. Instead, all belong to ectothermic tetrapods (lissamphibians and snakes; Rage, 2006) and to arthropods (Schwermann et al., 2016a; Schwermann et al., 2016b). Might this result from a taphonomic filter? Was the environment in which these fossils formed (only for the lost locality that yielded mummies) more suitable for lissamphibians and snakes than for mammals? The fact that this locality is now lost prevents us from answering the questions raised above, for now. However, Schwermann et al. (2016a) suggested that such fossils (i.e., arthropods in that case) formed by rapid permineralization of phosphate transported by water that circulated in the fissures and fillings. They suggested that the source of phosphate might have been the numerous bones that accumulated in the fissures. However, another origin of vertebrate mummies deserves consideration. Bats are very numerous in the localities of the Phosphorites (Sigé & Hugueney, 2006) and they likely produced a large amount of guano. Bat guano, which is very rich in phosphate, is known to facilitate preservation in the presence of calcite (Shahack-Gross et al., 2004). Permineralization of soft tissues by phosphorus leading to exceptional preservation was already observed in a few other cases, for embryophytes, arthropods and gastropods (Arena, 2008), ostracod sperm (Matzke-Karasz et al., 2014), and annelids (Wilson et al., 2016). Schwermann et al. (2016a) also showed that air-dried specimens (as can be observed nowadays in lissamphibians after post-mortem desiccation) do not accurately preserve soft tissues. This suggests that dead animals were rapidly buried in the sediment, a prerequisite for phosphatization of soft tissues (Wilson et al., 2016), where they were infiltrated by percolating water and thus permineralized. In any case, given the amazing three-dimensional preservation of soft tissues, we believe that it is appropriate to classify the lost locality of the ‘Phosphorites du Quercy’ that produced the vertebrate mummies (and the locality that yielded the arthropod mummies) as a Fossil Konservat-Lagerstätte.

Conclusions

The only specimen of Phosphotriton sigei represents a peculiar case of exceptional preservation, in which several organs are preserved in three dimensions, in addition to the skeleton: lung, spinal cord, lumbosacral plexus, digestive tract, muscles, and an unidentified urogenital organ. In addition, the alimentary tract contains skeletal remains of a frog, which is a very rare prey for salamanders. Contrary to the above-cited case of arthropods (Schwermann et al., 2016a), we do not believe that the new data on soft anatomy will revolutionize our understanding of lissamphibian evolution, particularly because such characters have played a modest role in phylogenetic studies of lissamphibians. However, these data, such as the presence of a lung, proved critical to place the mummy in the phylogeny, and these data document the oldest known occurrence of anurophagy in urodeles.

Supplemental Information

Supplemental Information 1 3D model of the skeleton of MNHN.F.QU17755, holotype of Phosphotriton sigei

File should be open with Adobe Acrobat Reader for 3D content. Unit for measurement is in cm.

Click here for additional data file.

Supplemental Information 2 Results of Bone Profiler analyses for bone microanatomy of the femur of MNHN.F.QU17755

Blue line of the graph is radial profile; red line is global profile.

Click here for additional data file.

We thank an anonymous reviewer and Pavel P. Skutschas for their comments which helped us improve our manuscript. We thank Patrick Orr for suggesting helpful references during the writing of the manuscript. We are grateful to Renaud Boistel and Vincent Fernandez for their help in the acquisiton of the data. We thank the staff of the Centre de microscopie de Fluorescence et d’IMagerie numérique (CeMIM) facilities of the MNHN, and particularly Marc Gèze and Cyril Willig for providing access to computers during the segmentation of the data. The synchrotron microtomography experiments were performed on the ID19 (proposal MD727) beamline at the European Synchrotron Radiation Facility (ESRF), Grenoble, France.

Additional Information and Declarations

Competing Interests

Author Contributions

Data Availability

The authors declare there are no competing interests.

Jérémy Tissier performed the experiments, analyzed the data, wrote the paper, prepared figures and/or tables, reviewed drafts of the paper.

Jean-Claude Rage and Michel Laurin conceived and designed the experiments, analyzed the data, reviewed drafts of the paper.

The following information was supplied regarding data availability:

The 3D model of the holotype of Phosphotriton sigei, in 3D PDF file format, has been provided as a Supplemental File.

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
