# Peer review of "Exceptional soft tissues preservation in a mummified frog-eating Eocene salamander"

_PeerJ, doi:10.7717/peerj.3861_

## Round 0.1 · original submission · Minor Revisions

Both reviewers consider that this is an interesting manuscript, and I agree with them. Please, take their suggestions in full consideration so that we can proceed with the publication process.

Reviewer 1 ·

Basic reporting

The report is written mainly clear English, but see below.
Literature is appropriate.
Figs are excellent.
No clear hypotheses but not necessary in this report. Considerable speculation but generally ok with me.

Experimental design

Not relevant

Validity of the findings

I think the descriptive information and especially the visuals are of high quality.

Additional comments

I would tuck the information gathered from this manuscript in that part of my brain labeled “Cabinet of Curiosities”. The phosphorized fossil is remarkably well preserved and the authors have derived a lot of information from the scans. I am interested in salamander anatomy, and accordingly enjoyed reading about there methods and interpretations.
What I take home from this manuscript is that under certain circumstances, not well understood, amazing preservation of anatomical detail is possible in fossils. That is it. It is not known exactly where the fossil was found and so taphonomic details are lacking and speculation results.
Line 69 is an incomplete sentence and the entire complex sentence would be better as several sentences. Technical English is best in short declarative sentences.
I think the anatomical inferences are generally good and justified. I think the authors are correct regarding muscles, lung, cloacal glands and even the partial frog in the gut.
The frog in the gut of a relatively small salamander is a surprise. Salamanders generally do not eat frogs or even tadpoles (I can add Ambystoma gracile to the short list of salamanders known to eat metamorphosing frogs). In this instance the frog seems to be partially dismembered, or has been digested in a peculiarly selective manner).
Even without the lung evidence, this is not a plethodontid on the basis of vertebral anatomy and especially the very long and well-developed ribs.

·

Basic reporting

No comments

Experimental design

No comments

Validity of the findings

The paper is a valuable contribution and I strongly support its publication in the PeerJ (after very minor revision).

Additional comments

I have no major suggestions on the paper. I enjoyed the opportunity to read this MS and I look forward to see it published. My minor suggestions for the improvement of the MS are listed below:
“Abstract”, line 26, I recommend to add “juvenile” before “frog”. It is important to provide a characteristic of a prey (= the swallowed frog was juvenile).

“Introduction”, line 47. Please, clarify why “external features” are “notable” (e.g. notable for taxonomic identification?)

“Results”, “Lung”, line 156. I recommend to remove the phrase “(i.e. within the rib cage)” because salamanders do not have a rib cage (this structure is typical for amniotes)

“Discussion”, “Exceptional preservation”, line 193, I recommend to use the name Vertebrata (Latin) or vertebrates (English), but not Vertebrates.

I waive all anonymity for my review. The authors are welcome to contact me directly should they have questions or anything they wish to discuss.

Pavel Skutschas

---

## Round 0.2 · accepted · Accept

Thank you for your consideration of all suggestions of the reviewer. The ms is ready for publication.